# Extracellular PKM2 Preserves Cardiomyocytes and Reduces Cardiac Fibrosis During Myocardial Infarction

**DOI:** 10.3390/ijms252413246

**Published:** 2024-12-10

**Authors:** Yang Huang, Bin Li, Zongxiang Gui, Erhe Gao, Yi Yuan, Jenny Yang, Khan Hekmatyar, Falguni Mishra, Payton Chan, Zhiren Liu

**Affiliations:** 1Department of Biology, Georgia State University, Atlanta, GA 30303, USA; yh344@georgetown.edu (Y.H.); bli17@student.gsu.edu (B.L.); yyuan13@student.gsu (Y.Y.); fmishra1@student.gsu.edu (F.M.); pchan2@student.gsu.edu (P.C.); 2Department of Chemistry, Georgia State University, Atlanta, GA 30303, USA; zgui1@gsu.edu (Z.G.); jenny@gsu.edu (J.Y.); nhekmatyar@gsu.edu (K.H.); 3Center for Translational Medicine, Temple University, Philadelphia, PA 19140, USA; erhe.gao@temple.edu

**Keywords:** pyruvate kinase M2, cardiomyocytes, myocardial infarction, integrin α_v_β_3_

## Abstract

Substantial loss of cardiomyocytes during heart attacks and onset of other cardiovascular diseases is a major cause of mortality. Preservation of cardiomyocytes during cardiac injury would be the most effective strategy to manage these diseases in clinic. However, there is no effective treatment strategy that is able to prevent cardiomyocyte loss. We demonstrate here that the systemic administration of a recombinant PKM2 mutant (G415R) preserves cardiomyocytes and reduces cardiac fibrosis during myocardial infarction. G415R preserves cardiomyocytes by protecting the cardiomyocytes from dying and by promoting cardiomyocyte proliferation. Preservation of cardiomyocytes by extracellular PKM2 (EcPKM2) reduces cardiac fibrosis because of the decreased activation of cardiac fibroblasts. Our experiments show that EcPKM2 (G415R) exerts its action by interacting with integrin a_v_b_3_ on cardiomyocytes. EcPKM2(G415R) activates the integrin–FAK–PI3K signaling axis, which subsequently suppresses PTEN expression and consequently regulates cardiomyocyte apoptosis resistance and proliferation under hypoxia and oxidative stress conditions. Our studies uncover an important cardiomyocyte protection mechanism. More importantly, the activity/action of EcPKM2 (G415R) in preserving cardiomyocyte suggesting a possible therapeutic strategy and target for the treatment of heart attacks and other cardiovascular diseases.

## 1. Introduction

Cardiomyocytes are the major cell type in the adult heart that maintain the heart function of pumping blood to the whole body. Loss of cardiomyocytes due to myocardial infarction is the most important cause for morbidity and mortality in heart diseases. The acute loss of cardiomyocytes caused by myocardial infarction cannot be replaced due to the very limited regenerative capability of adult cardiomyocytes [1,2,3,4,5]. Myocardial infarction leads to large-area ischemia. Hypoxia resulting from ischemia induces necrotic death of cardiomyocytes. Subsequent reperfusion leads to destructive inflammation and oxidative stress, which results in further cardiomyocyte death by various mechanisms [5,6,7]. Preservation of cardiomyocytes is critically important to sustain patient survival after suffering from a heart attack. Importantly, the death of cardiomyocytes quickly turns on local tissue repair mechanisms with the activation of cardiac fibroblasts. Activated cardiac fibroblasts release the extracellular matrix (ECM), mostly collagen, to form scar tissue in order to prevent rupture of the myocardium immediately after infarction. However, the continuous accumulation of ECM, particularly collagen, disrupts normal myocardium functions, which often lead to heart failure [8,9,10]. It is generally believed that damaged myocardium tissues due to the death of cardiomyocytes are re-filled or replaced by ECM that is released from activated cardiac fibroblasts [9,11]. There are close communications between dying cardiomyocytes and cardiac fibroblasts to control fibroblast activation during myocardial infarction [10], including: (1) The dying cardiomyocytes communicate with cardiac fibroblasts by paracrine signaling via releasing cytokines, hormones, and growth factors; and (2) they also crosstalk via a Gap-junction mainly for electronic coupling [6,10]. There is substantial evidence to support that dying cardiomyocytes are the main trigger for the activation of cardiac fibroblasts, which consequently results in cardiac fibrosis. It is plausible that preserving cardiomyocytes during myocardial infarction could avoid the excessive activation of cardiac fibroblasts and thus avoid the accumulation of ECM/collagen in the myocardium tissues, consequently preventing cardiac fibrosis.

Pyruvate kinase is an enzyme that catalyzes the last reaction in glycolysis. There are four isoforms of pyruvate kinases, L/R and M1/M2, which are expressed in different tissue types or under different physiological conditions [12,13]. PKM2 can form a homodimer or a homotetramer. The tetramer is active as pyruvate kinase [14,15], while the dimer is a protein kinase [16,17,18]. Several PKM2 mutants favor the dimer form, including a G415R mutant that had been derived from a cancer patient. G415R exists as a dimer with high solubility in solution [19,20]. Interestingly, a number of recent studies show that PKM2 is functionally involved in multiple cellular processes in different locations, including metabolism control, transcription regulation, and chromatin packaging [17,21,22]. High serum levels of PKM2 have long been observed in patients with various inflammation diseases [23,24,25], indicating a potential association of EcPKM2 in inflammatory responses. It has also been shown in several laboratories that PKM2 is functionally involved in several non-metabolic cellular processes in various cell types [26,27]. These studies suggested the potentially important functional role of PKM2 in inflammatory responses to tissue injury and cancer progression. Our previous studies demonstrated that EcPKM2 facilitates tumor growth by promoting angiogenesis [28]. During cutaneous wound repair, PKM2 is released by infiltration neutrophils at the wound site. The released PKM2 facilitates wound repair [29]. Interestingly, PKM2 is expressed in cardiomyocytes after the fourth day of myocardial infarction. PKM2 in cardiomyocytes promotes cell cycle progression, thus facilitating cardiomyocyte proliferation and regeneration [30]. We report here that the systemic administration of a recombinant PKM2 mutant (G415R), which mostly adopts a dimer form, preserves cardiomyocytes and decreases cardiac fibrosis during myocardial infarction. The administration of G415R prevents cardiomyocyte death and facilitates cardiomyocyte proliferation under hypoxia and oxidative stress conditions. The PKM2 mutant interacts with integrin α_v_β_3_ on cardiomyocytes in order to activate the FAK–PI3K signaling axis, which suppresses PTEN expression downstream and consequently promotes survival and proliferation. Consistent with previous reports [30], we show that PKM2 is expressed and released into the extracellular space 4 days post myocardial infarction in the mouse model. PKM2 expression and extracellular PKM2 are detected in the myocardium tissues of infarction patients and the infarcted mouse model. Also consistent with previous reports, integrin α_v_β_3_ is expressed in the myocardium tissues of infarction patients and the infarcted mouse model. The integrin is expressed as early as 6 h after myocardial infarction. Our study provides a potential therapeutic strategy for heart attack treatment.

## 2. Results

### 2.1. PKM2 Is Expressed in and Secreted from Cardiomyocytes Under Stress Conditions

Ajit Magadum and co-workers reported that PKM2 is expressed in neonatal cardiomyocytes and adult cardiomyocytes after myocardial infarction induction. However, the expression of PKM2 in the infarcted myocardium tissues of adult cardiomyocytes only occurs 4 days after infarction [30]. This observation prompted us to examine PKM2 expression and secretion in the myocardium tissues of infarcted hearts. First, we carried out immunohistochemistry (IHC) analysis of PKM2 in tissue samples from myocardial infarction patients. PKM2 was expressed in the infarcted heart tissues, while PKM1 staining was detected in normal healthy cardiac tissue. Extracellular PKM2 was obvious in the magnified images of staining of the myocardium tissues of infarction patients (Figure 1A, magnified region to show EcPKM2). We also performed IHC staining of PKM2 in the myocardium tissues of infarcted mice after different time points of infarction. No PKM2 staining was observed 6 h, 24 h, and 3 days after infarction induction. PKM2 staining was clearly observed in the infarction regions 4 days after infarction induction. EcPKM2 was clearly visible in the IHC staining (Figure 1B,C, magnified region to show EcPKM2). This is consistent with the observation that PKM2 is present in the myocardium tissues of infarcted hearts 4 days after infarction [30]. To verify the expression and secretion of PKM2 from cardiomyocytes, we examined PKM2 in a culture medium of commercially available adult human primary cardiomyocytes under hypoxia and normoxia conditions by ELISA. The authenticity of the primary cardiomyocyte was demonstrated by the expression of cTnT in the cells (Appendix A). PKM2 was not detected in the medium under normoxia conditions. However, PKM2 was detected in the culture medium under hypoxia conditions (Figure 1D). If PKM2 is expressed in the infarcted heart and is released into the extracellular space, we anticipate that PKM2 would be detectable in the blood circulation of the infarcted mice. We performed ELISA assays of PKM2 in plasma samples from mice after different time points of myocardial infarction. Consistent with what we had anticipated, PKM2 was not detected in the plasma of the sham mice or in the plasma of the experiment mice 6 and 24 h post myocardial infarction induction. However, PKM2 was detected in the plasma of the experiment mice 7 days after the infarction induction (Figure 1E). The experiments from other laboratories, as well as our own, suggest that PKM2 is expressed and released into the extracellular space ~4 days after myocardial infarction.

### 2.2. EcPKM2 Protects Cardiomyocytes from Apoptosis and Promotes Proliferation

We previously reported that EcPKM2 protected myofibroblasts from apoptosis by interacting with integrin α_v_β_3_ and subsequently activating integrin signaling [31]. Furthermore, EcPKM2 facilitated wound repair and tissue regeneration [28,29]. Independent studies in other laboratories have also demonstrated that integrin α_v_β_3_ is expressed in cardiomyocytes under cardiac infarction conditions [32,33,34,35,36]. Thus, we reasoned that EcPKM2 may have a function to protect cardiomyocytes from apoptosis by interacting with the integrin and activating the integrin signaling. We first probed the expression of the integrin α_v_β_3_ in the H9C2 cells by immunofluorescence (IF) staining and immunoblot. Clearly, the integrin is expressed in the cells under hypoxia conditions but not under normoxia conditions (Appendix A). We next analyzed the integrin expression in the human primary cardiomyocytes under hypoxia and oxidative stress conditions. Evidently, the integrin is expressed in cardiomyocytes under the stress conditions, while the integrin is not expressed under normoxia conditions (Figure 1F,G). We then analyzed the integrin expression in the myocardium tissues of infarcted mice by IHC staining. Clearly, integrin β3 was detected in the myocardium tissues of the infarction regions. Integrin β_3_ started to express within 6 h after ischemic injury and reached its highest level on day four, and the expression decreased nine days post infarction induction. The integrin was not detected in non-infarction areas (Figure 1H,I). To confirm the clinical relevance of the integrin expression, we analyzed the integrin expression in the heart tissues of myocardial infarction patients. It is clear that the integrin is highly expressed in the infarction regions (particularly border zones), while it is almost undetectable in the normal non-infarction regions (Appendix A). We therefore conclude that integrin α_v_β_3_ is upregulated in cardiomyocytes under hypoxia and oxidative stress conditions.

To test whether EcPKM2 protects cardiomyocytes from apoptosis under stress conditions, we employed a recombinant PKM2 mutant (referred to as G415R) that exists mostly as dimer with high solubility in solution [20,37], while recombinant PKM1 (rPKM1) was used as control. G415R exhibited higher solubility and better stability compared with the recombinant wt. PKM2. G415R not only protected human primary cardiomyocytes from apoptosis under hypoxia conditions (Figure 2A,B) but also promoted human primary cardiomyocyte proliferation under hypoxia conditions (Figure 2C–E). We then tested if G415R had similar effects on the H9C2 cells. Clearly, G415R protected the H9C2 cells from apoptosis under hypoxia and oxidative stress conditions (Appendix A). Similarly, G415R also promoted cell proliferation of the H9C2 cells under hypoxia and oxidative stress conditions (Appendix A).

### 2.3. EcPKM2 Preserves Cardiomyocytes During Myocardial Infarction

Protection of cardiomyocytes from apoptosis and facilitation of proliferation under hypoxia and oxidative stress conditions suggest a potential role in heart myocardium protection during myocardial infarction. We therefore examined the effects of the G415R PKM2 mutant on myocardial infarction using the commonly used mouse model of left coronary artery ligation (MI) and ischemia–reperfusion (IR) [38]. Animals in the MI groups were pretreated with one dose of G415R 24 h before the artery ligation. The day, the animals received either the MI or the IR myocardial infarction induction. Both the MI and the IR mice were then continuously treated with G415R (twice weekly). The experimental animals were euthanized at designated time points for analyses (as shown in Figure 2F). His-tag IHC staining demonstrated the present of recombinant G415R in the myocardium tissues of the infarction regions (Appendix A). MR imaging showed success in the creation of myocardial infarction due to the MI and IR artery ligation processes (Appendix A). The administration of G415R decreased MI mouse death (Figure 2G). The administration of G415R also decreased the heart weight in both the MI and the IR mice 30 days after infarction (Appendix A). MR imaging analyses of blood flow in the heart suggested that G415R treatment improved the blood flow and restored blood flow of the MI and IR mice almost to the same level of the sham mice (Appendix A). MR images also revealed that G415R led to thicker left ventricle wall compared with those of vehicle- and PKM1-treated groups. Masson’s trichrome staining of the myocardium tissues of infarcted hearts showed that mice treated with G415R had smaller infarction scar sizes compared with the rPKM1- and vehicle-treated groups in both the MI and the IR mice. There was almost no myocardium remodeling 30 days post infarction in both the MI and the IR G415R-treated mice. Close examination of the magnified images revealed that G415R largely preserved myocardium structure similar to that of the sham mice in both the MI and the IR infarcted mice (Figure 3A–F and Appendix A). The effects of G415R on myocardial infarction sizes were further confirmed by wheat germ agglutinin (WGA) staining of the infarcted myocardium tissues (Appendix A). Close examination of the magnified WGA staining images revealed that G415R reduced the cardiomyocyte cross-sectional areas to the levels similar to that of the sham groups, while the rPKM1 and vehicle groups had larger cardiomyocyte cross-sectional areas in both the MI and the IR mice (Figure 3G–H and Appendix A), suggesting that EcPKM2 reduced myocardial hypertrophy during myocardial infarction. To explore whether G415R protects cardiomyocyte from apoptosis, TUNEL staining was performed with myocardium tissues of the infarction regions of mice hearts 6, 24, and 168 h after the infarction induction. Evidently, G415R decreased cardiomyocyte apoptosis at 6 and 24 h with both the MI and the IR models (Figure 4A–E and Appendix A). G415R had limited benefit on prevention of cell death for cardiomyocytes 7 days after infarction (Figure 4E and Appendix A). Cardiomyocyte apoptosis was not detected by TUNEL staining with myocardium tissues 30 days after infarction induction. Because G415R stimulates the human primary cardiomyocyte and H9C2 cell proliferation under hypoxia and oxidative stress conditions, we therefore examined whether G415R also facilitated cardiomyocyte proliferation in the myocardium tissues of infarcted hearts by Ki67 IHC staining. G415R treatment did not lead to cardiomyocyte proliferation 6 h after infarction induction. However, cardiomyocyte proliferation was apparent in the myocardium tissues of both the MI and the IR mice that were treated with G415R 24 h and 7 days after the infarction induction (Appendix A). The effects of G415R on cardiomyocyte proliferation were almost non-observable 30 days after infarction induction (both MI and IR), indicating that the time window for promoting cardiomyocyte proliferation/regeneration is between 1 and 7 days post infarction induction. To verify the proliferation of cardiomyocytes in the infarction regions of infarcted hearts, we carried out co-staining with IHC Ki67 and cardiac troponin I (cTnI), a molecule that is specifically expressed in cardiomyocytes. Proliferation of cardiomyocytes was observed with the Ki67 and cTnI co-staining of the myocardium tissues of G415R-treated infarcted mice. However, there was almost no co-staining with the vehicle- and rPKM1-treated mice (Appendix A). Death of cardiomyocyte results in the release of cardiac troponin (cTn) into the blood circulation, which is an important circulation molecular marker in reflecting cardiac damage. We therefore measured cTnI in blood collected from the infarcted mice. Clearly, almost no cTnI was detected in the plasma of the sham mice. The cTnI levels reached their highest levels 24 h post infarction. However, G415R treatment substantially reduced the cTnI levels in the blood circulation of the infarcted mice 24 h and 7 days post infarction (Figure 4F). Taken together, our experiments suggest that the systemic administration of the recombinant PKM2, e.g., G415R mutant, to the MI and IR mice prevented cardiomyocyte death and promoted cardiomyocyte proliferation during infarction.

### 2.4. EcPKM2 Reduces Cardiac Fibrosis During Infarction

Loss of cardiomyocytes due to infarction triggers the activation of cardiac fibroblasts to release ECM/collagen to repair the damaged myocardium. The persistent activation of cardiac fibroblasts leads to an accumulation of ECM in the myocardium tissues and to cardiac fibrosis [11,39]. We previously reported that EcPKM2 facilitates fibrosis progression by protecting myofibroblasts from apoptosis in rodent liver and lung fibrosis models [31]. We questioned whether the systemic administration of G415R would facilitate cardiac fibrosis progression during myocardial infarction by acting on cardiac fibroblasts. Therefore, we analyzed cardiac fibers in the Masson’s trichrome staining of the myocardium tissues in the infarction regions of infarcted hearts. The G415R-treated mice had much less collagen accumulation in the infarction regions in both the MI and the IR models compared with the rPKM1- and vehicle-treated groups (Appendix A). The results were further confirmed by analyses of wheat germ agglutinin (WGA) staining of the infarcted myocardium tissues (see Figure 3A,B,G,H and Appendix A). A likely explanation for the accumulation of less ECM/collagen fibers in the myocardium tissues of the G415R-treated mice is that G415R treatment might lead to the activation of fewer cardiac fibroblasts in the MI and IR mice due to preservation of cardiomyocytes. We examined the activation of cardiac fibroblasts in the infarcted myocardium tissues of the MI and IR mice by IHC staining of α-SMA, a molecular marker of myofibroblasts. G416R treatment led to substantially less staining of α-SMA in the myocardium tissues of the MI and IR mice 7 and 30 days post infarction induction (Figure 4G–I and Appendix A).

### 2.5. EcPKM2 Interacts with Integrin α_v_β_3_ and Activates the Integrin and Downstream Signaling in Cardiomyocytes Under Stress Conditions

Cardiomyocytes express integrin α_v_β_3_. EcPKM2 protects cardiomyocytes from apoptosis. As we demonstrated previously, EcPKM2 protects myofibroblasts from apoptosis by interacting with the integrin and activating the integrin signaling [31], and we therefore wondered whether EcPKM2 acted on cardiomyocytes by a similar mechanism. To answer this question, we first carried out co-immunoprecipitation of G415R with extracts prepared from the primary human cardiomyocytes under hypoxia conditions. Evidently, G415R co-precipitated with integrin β3, as assayed using both antibodies against PKM2 or integrin β3 as a pull-down antibody (Figure 5A,B). To analyze whether EcPKM2 exerts its bioaction on cardiomyocytes by activating integrin signaling, we first examined the activation of integrin signaling. An increase in the activation of FAK was observed upon the addition of G415R to the culture medium of human primary cardiomyocytes under hypoxia conditions. G415R had no effect on FAK under normoxia conditions. The activation of FAK was reduced by the antibody LM609, an antibody binds to and blocks integrin α_v_β_3_ at the ligand site, and the antibody IgGPK (Figure 5C–F). Similarly, the effects of G415R on the activation of FAK were observed with the H9C2 cells (Appendix A). It is well established that the activation of integrin signaling leads to the activation of FAK and subsequent downstream PI3K [40]. Thus, we examined the activation of PI3K in the human primary cardiomyocytes and the H9C2 cells upon G415R treatments. Immunoblot and PI3K activity assays demonstrated that PI3K was upregulated (Figure 5C–F and Appendix A) and activated in the cells (Figure 5G). Furthermore, the activation of PI3K was abrogated by a FAK inhibitor and a PI3K inhibitor (Appendix A). Consistent with the observed effects of G415R on the protection of cardiomyocytes from apoptosis, G415R decreased the cleaved PARP and caspase 3 in the treated human primary cardiomyocytes (Figure 5H,I). If protection from apoptosis and promotion of proliferation of cardiomyocytes by EcPKM2 is mediated by PI3K activation, the effects would be abolished by a PI3K inhibitor. As expected, the effects of G415R on apoptosis protection and proliferation of the H9C2 cells were abolished by commercially available FAK and PI3K inhibitor (Appendix A). Our results support the conclusion that EcPKM2 interacts with integrin α_v_β_3_ and consequently activates the integrin signaling in cardiomyocytes under hypoxia and oxidative stress conditions, thus protecting cardiomyocytes from apoptosis and promoting proliferation.

PI3K–PTEN pathway regulates cardiomyocyte apoptosis and proliferation [41,42]. Reduction in expression or pharmacological inhibition of PTEN in cardiomyocyte prevents cardiomyocytes death and promotes proliferation [41,43,44,45]. We show that EcPKM2 downregulates PTEN in myofibroblasts via interaction with integrin α_v_β_3_ and the activation of the FAK–PI3K signaling axis [31]. It is also demonstrated that integrin α_v_β_3_ signaling promotes proliferation and attenuates hypoxia-induced apoptosis via PTEN/Akt/mTOR signaling [32]. Thus, we wondered whether EcPKM2 regulates cardiomyocyte apoptosis and proliferation under hypoxia and oxidative stress conditions by the same FAK–PI3K signaling axis via reduction of PTEN in cardiomyocyte. We first probed the PTEN levels and activity in the G415R-treated H9C2 cells under hypoxia and normoxia conditions. PTEN was not expressed in high levels in the cells under normoxia conditions. The addition of G415R into the cell culture had only marginal effects on PTEN expression. PTEN was highly expressed in the cells under hypoxia conditions. The addition of G415R (but not rPKM1 or vehicle) into the culture medium reduced PTEN in the cells (Figure 6A,B), suggesting that EcPKM2 might play a role in downregulation of PTEN in cardiomyocyte during myocardial infarction, which subsequently protects cardiomyocytes from apoptosis and promotes proliferation. To further verify the effects of EcPKM2 on PTEN in cardiomyocytes, we analyzed the PTEN levels in the infarction regions of the myocardium tissues of the infarcted mice by IHC. Evidently, there was only residual PTEN expression in the sham mice. PTEN was elevated in both the MI and the IR mice. G415R treatment reduced PTEN by ~3 folds (Figure 6C,D). The observation supports the conclusion that EcPKM2 activates the integrin α_v_β_3_–FAK–PI3K signaling axis, which consequently reduces PTEN in cardiomyocytes during infarction.

## 3. Discussion

Our study showed an important function of EcPKM2 in the preservation of cardiomyocytes during myocardial infarction, which demonstrates an exciting potential application of the administration of recombinant PKM2 (e.g., the dimer mutant) as a treatment agent for heart attacks and other cardiovascular diseases. The role of the administration of rPKM2 in protecting cardiomyocyte from apoptosis and promoting cardiomyocyte proliferation is in concordance with our previous observations: (1) EcPKM2 facilitates tumor growth [28] and cutaneous wound healing [29]; and (2) EcPKM2 facilitates organ tissue fibrosis progression by promoting myofibroblasts apoptosis resistance and upregulating Arg-1 expression via activating integrin α_v_β_3_. The studies clearly support the theme that EcPKM2 may bridge the early inflammatory response to a later proliferation phase (fibrogenesis/angiogenesis) during tissue injury repair/regeneration.

Cardiomyocytes start to express integrin α_v_β_3_ early in myocardial infarction. EcPKM2 interacts with the integrin on cardiomyocytes under stress conditions, which subsequently activates the FAK–PI3K signaling axis and suppresses PTEN expression. The action of EcPKM2 leads to a reduction in cardiomyocyte apoptosis and an increase in proliferation. Preservation of cardiomyocytes by the administration of EcPKM2 results in the activation of fewer cardiac fibroblasts and, thus, lessens the chance of cardiac fibrosis (Figure 6E). PKM2 is expressed and released from cardiomyocytes after myocardial infarction, suggesting a possible intrinsic protection or repair mechanism. However, the expression and release of PKM2 into the extracellular space come too late in protection of myocardium during infarction. It has been demonstrated that forcing an early expression of PKM2 in cardiomyocytes during infarction may enable a protection role [30]. We previously also observed that PKM2 is released into the extracellular space 3 days after introducing a cutaneous wound. Similarly, expression and release of PKM2 come late in protection and facilitation of repair. It is conceivable that exogenously introducing PKM2 earlier during tissue injury would facilitate protection and repair. Our experiment demonstrates that cardiomyocytes express and secrete PKM2 under stress conditions. Can PKM2 be released from other source cells in the myocardium tissues during infarction is an open question. Our previous reports demonstrated that PKM2 can be released from neutrophils and myofibroblasts [29,31].

Based on our previous observations of the action of EcPKM2 on myofibroblasts, it was unexpected that EcPKM2 reduced cardiac fibrosis after infarction [31]. One likely explanation is that preservation of the cardiomyocytes reduced cardiac myofibroblast activation/differentiation. It is generally believed that damaged myocardium tissues due to the death of cardiomyocytes are re-filled or replaced by ECM that is released from activated cardiac fibroblasts [46,47]. There are close communications between dying cardiomyocytes and cardiac fibroblasts to control fibroblast activation during myocardial infarction [6,48,49]. There is substantial evidence to support the notion that dying cardiomyocytes are the main trigger for the activation of cardiac fibroblasts, which consequently results in cardiac fibrosis. G415R treatment results in substantially fewer cardiomyocyte deaths, which consequently leads to a lesser degree of activation of the cardiac fibroblasts in the infarction regions (as shown Figure 4G–I). However, alternative possibilities cannot be excluded. For example, EcPKM2 may alter the immune cell or cytokine profiles in the infarction regions, which consequently controls cardiac fibroblast activation.

## 4. Materials and Methods

### 4.1. Recombinant Protein Expression and Purification

The cDNAs that encode human G415R and PKM1 were purchased from Adgenes. The cDNAs were subcloned into bacterial expression vectors. The recombinant proteins were expressed and purified from bacterial lysates by a two-column procedure as in previous reports [17,28]. 

Patient tissue analyses were carried out under the guidelines of NIH and adhered to the Declaration of Helsinki. The study was approved by GSU Institutional Review Board (IRB), and it falls under IRB exemption 4. All tissue samples were de-identified, and the samples were sectioned and analyzed by different staining methods. Diseased and normal heart tissue samples were purchased from US Biomax Inc. (Rockville, MD, USA). 

### 4.2. Animal Model and Treatment

All animal experiments conformed to the guidelines of the NIH Guide for the Care and Use of Laboratory Animals and were approved by Georgia State University IACUC. At the end of each experiment, the mice were euthanized by CO_2_ inhalation. Animal euthanasia conformed to the guidelines of the NIH Guide for the Care and Use of Laboratory Animals. Male C57BL/6J mice aged 11–12 weeks were used. The mice were anesthetized with 2% isoflurane inhalation without ventilation. A small skin cut (1.2 cm) was made over the left chest, and a purse-string suture was made around the cut. After dissection and retraction of the pectoral major and minor muscle, the fourth intercostal space was exposed. A small hole was made at the fourth intercostal space with a mosquito clamp to open the pleural membrane and pericardium. The heart was gently squeezed out of the hole with the clamp slightly open. The LCA was located, sutured, and ligated at a site 2–3 mm from its origin using a 6–0 silk suture. The ligation was successful when the anterior wall of the LV turned pale. After ligation, the heart was immediately placed back into the intrathoracic space, followed by a gentle squeezing the chest to remove the air and closing of the muscle and the skin by means of the previously placed purse-string suture. The mouse was then allowed to breathe room air and monitored during the recovery period, which was generally complete within 3 to 5 min. The sham group underwent the same surgical procedure, except that the LCA was not occluded. 

The IR injury procedure in mice is essentially the same as inducing the MI injury, except that a slipknot was tied around the LCA 2 to 3 mm from its origin with a 6–0 silk suture. The heart was then quickly placed back into the thoracic space, followed by manual evacuation of air and the skin closing. The mouse was re-anesthetized with 2% isoflurane inhalation, the chest reopened, and the slipknot was released by pulling the long end of the slipknot suture smoothly and gently, followed by manual evacuation of the pneumothorax and chest closure. 

Mice in the MI treatment were dosed 24 h before surgery via i.p. injection and again ~1 h before surgery. Mice in the IR treatment were dosed at the time of surgery. All animals were dosed twice per week for up to 2 weeks. At the end of the treatments, the animals were sacrificed, and the hearts were harvested and measured, followed by processing for embedding and sectioning. Statistical analyses were done in comparison with the control group.

### 4.3. Cell Culture

The H9C2 cells were maintained in a DMEM medium supplemented with 10% Fetal Bovine Serum (FBS) in 37 °C with 5% CO_2_. The human cardiomyocytes were purchased from SCIENCELL (Cat # 6200). The cardiomyocytes were isolated from adult human hearts by the vendor. The cardiomyocytes were maintained in the cardiac myocyte medium culture based on the vendor’s instructions. Culture flasks or dishes were coated with poly-L-lysine (2 μg/cm^2^) for human cardiomyocytes. For hypoxia experiments, all cells were starved in the medium without phenol red and FBS overnight and stimulated with G415R. Cells were then cultured in a hypoxia incubator chamber flushed with low oxygen gas (5% CO_2_, 1% O_2_, and 94 N_2_) for 24 or 48 h. For oxidative stress experiments, the cells were pre-incubated in the hypoxia incubator chamber for 24 h, and 300 μM of H_2_O_2_ was added to the cells for 5 h to introduce apoptosis. The inhibitors (20 μg/mL LY294002; 2.5 μM FI14) and antibodies (10 ng/mL LM609; 10 ng/mL PKM2 antibody) were added 30 min earlier than rPKM2/G415R addition for the respective experiments. After treatment, cells were stained or harvested at the indicated time points for the designated measurements and assays.

### 4.4. Flow Cytometry, Immunohistochemistry Staining, Immunofluorescent Staining, Masson’s Trichrome Staining, Immunoprecipitation, Immunoblot, ELISA, TUNEL Assay, Cell Viability Assay, Cell Proliferation Assay, PI3K Activity Assay

All the staining and assays were performed using commercially available staining kits or assay kits by the procedures as in our previous reports [31,50]. 

### 4.5. Wheat Germ Agglutinin (WGA) Staining

Wheat germ agglutinin (WGA) staining was performed using the commercially available WGA staining kits with conjugated Alexa Fluor probe.

### 4.6. Staining Quantification

Quantification of Masson’s trichrome, WGA, IHC, and IF staining (including patient tissues and tissues from the animal experiments) was carried out using ImageJ (v1.53q 30 March 2022). Quantifications are positive stain areas in each view field or fold change compared with the controls, unless otherwise specified in the figures and legends. All quantification results were means of randomly selected 3 view fields per section, 5 sections per animal, and 5–10 mice per experimental group, unless otherwise specified in the figures and legends.

### 4.7. Cardiac Magnetic Resonance Imaging

Cardiac magnetic resonance imaging (CMR) examinations were performed 30 days after induction of acute myocardial infarction/myocardial ischemia–reperfusion injury. Imaging was performed on a 7 T preclinical scanner (Bruker, BioSpec 70/20 USR, Paravision 360, Billerica, MA, USA) using a transmit–receive volume coil (400 W maximum transmit pulse of 5 ms, inner diameter of 40 mm). Anesthesia was adjusted to maintain a respiration rate of 65 ± 5 breaths per minute. The temperature of the animals was kept at 35–37 °C by using a warm pad. Breathing and body temperature were monitored during CMR (SA Instruments, Inc., Stony Brook, NY, USA). Pre-contrast images in standard long-axis geometries (two-, three- and four-chamber view), which were used to plan the short-axis orientation covering the entire left ventricle (LV). A Bright-blood Intra Gate Fast Low Angle Shot (Cine_brightblood_IG_FLASH) was used to acquire 4–10 coronal slices for functional assessment of the LV, with repetition time = 100 ms, echo time = 3.5 ms, number of repetitions = 1, oversampling = 230, flip angle = 45°, slice thickness = 1 mm, field of view= 25 × 25 mm^2^, matrix = 192 × 192, in-plane resolution = 0.13 × 0.13 mm^2^. For all cine IG scans, 45 cardiac phases were reconstructed. Dedicated software (Segment CMR, v3.2 Medviso, Sweden) was used for imaging analysis. The LV epicardial and endocardial borders were manually traced on the end-diastolic (ED) and end-systolic (ES) image of each slice. Left ventricle end-diastolic volume (LV-EDV), left ventricle end-systolic volume (LV-ESV), stroke volume (SV), and ejection fraction (EF) were automatically calculated (SV = EDV − ESV; EF = SV/EDV).

### 4.8. Data Analyses and Statistical Calculations

All statistical analyses were carried out using the GraphPad Prism 9.0 software. All experiments were carried out 3 times minimum. Comparisons were made by two-tailed Student’s *t*-test when comparing two experimental groups. For a two-group comparison of unequal variance, the two-tailed Welch’s *t*-test was used. For a comparison between more than two groups, the statistical analyses were performed using two-way ANOVA with Tukey’s test. The *p* values <0.05 are regarded as statistically significant. In all figures and tables, ns means *p* > 0.05 and statistically insignificant, * means *p* < 0.05, ** means *p* < 0.01, and *** means *p* < 0.001.

## Figures and Tables

**Figure 1 ijms-25-13246-f001:**
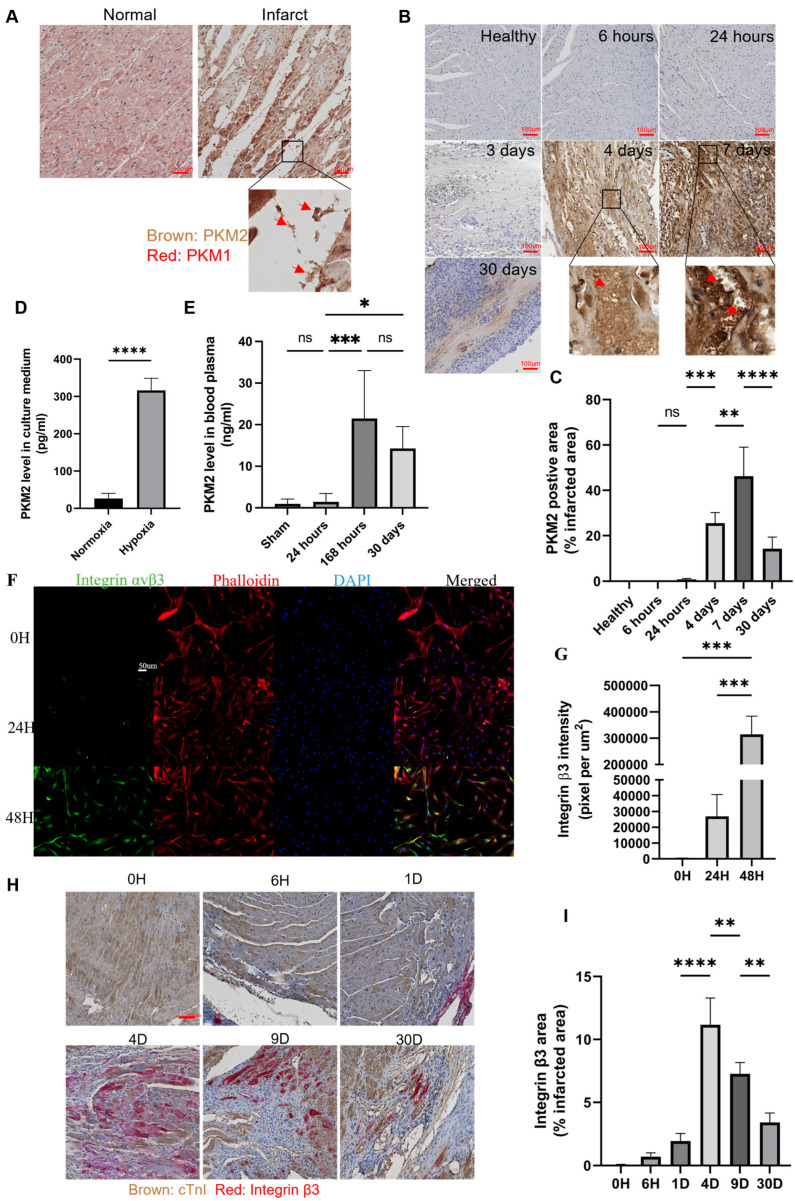
PKM2 is expressed and released into the extracellular space in the myocardium tissues 4 days after myocardial infarction. (**A**) Representative images (*n* = 6) of PKM1 (red) and PKM2 (brown) IHC staining of the myocardium tissues of the infarction regions of infarction patients. (**B**,**C**) Representative images (**B**) and quantification of PKM2 IHC staining ((**C**), *n* = 7) of myocardium tissues of the infarction regions of infarcted mice that were euthanized at the indicated time points after infarction induction. The quantity in (**C**) is presented as % of PKM2 positive area per view field (randomly selected 3 view fields per slide and 4 slides per animal), unpaired *t*-test. The arrows in the magnified images in (**A**,**B**) indicate extracellular PKM2. (**D**,**E**) ELISA analyses of PKM2 levels in the culture medium of primary human cardiomyocytes under normoxia and hypoxia conditions ((**D**), pg/mL, *n* = 6 unpaired *t*-test) and in plasma of infarcted mice at the indicated time points after infarction induction (**E**, ng/mL, *n* = 7 unpaired *t*-test). (**F**,**G**) Representative images (**F**) and quantification (**G**) of IF staining of integrin α_v_β_3_ in primary human cardiomyocytes cultured under normoxia or hypoxia conditions at different time points under hypoxia conditions. The quantity in (**G**) is presented as an integrin β3 staining intensity per view field, *n* = 5 unpaired *t*-test. (**H**,**I**) Representative images (**H**) and quantification (**I**) of IHC staining of integrin β_3_ in the myocardium tissues of infarcted mice at the indicated time points after infarction induction. The quantity in (**I**) is presented as integrin β3+ area as % of the infarction area (randomly selected 3 view fields per slide and 4 slides per animal), *n* = 7 unpaired *t*-test. Error bars in (**C**–**E**,**G**,**I**) represent mean ± S.E.M. Scale bars in (**A**,**B**,**H**) are 100 μm, in (F) 50 μm. ns—statistical non-significance, * *p* > 0.05, ** *p* > 0.01, *** *p* > 0.005, **** *p* > 0.001.

**Figure 2 ijms-25-13246-f002:**
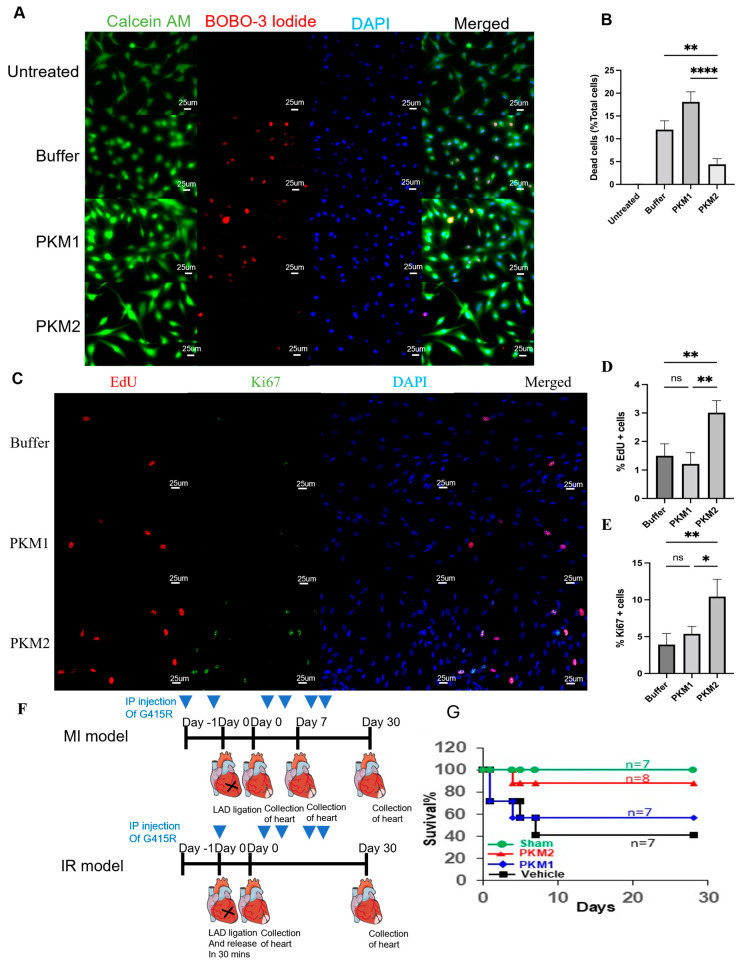
EcPKM2 prevents cardiomyocyte apoptosis and promotes proliferation under stress conditions. (**A**,**B**) Representative images (**A**) and quantification (**B**) of Calcein AM and iodide staining of human primary cardiomyocytes cultured under normoxia (untreated) and hypoxia conditions. The cultured cells were treated with the indicated agents. The quantity (**B**) is presented as % of dead cells in the total cell population, *n* = 5 unpaired *t*-test. (**C**–**E**) Representative images (**C**) and quantification (**D**,**E**) of EdU and Ki67 staining of human primary cardiomyocytes cells cultured under hypoxia conditions. The cultured cells were treated with the indicated agents. The quantities in (**D**,**E**) are presented as % of EdU (**D**) or Ki67 (**E**) positive cells in the total cell population, *n* = 5 unpaired *t*-test. (**F**) Schematical illustration of the MI and the IR animal experimental scheme. (**G**) Survival (%) of the MI mice after the indicated days of infarction induction and the indicated treatments. *n* is the group size. Sham is the group with surgery and without artery ligation or treatments. Error bars in (**B**,**D**) represent mean ± S.E.M. The scale bar in (**A**) is 25 μm and in (**C**) is 50 μm. ns—statistical non-significance, * *p* > 0.05, ** *p* > 0.01, **** *p* > 0.001.

**Figure 3 ijms-25-13246-f003:**
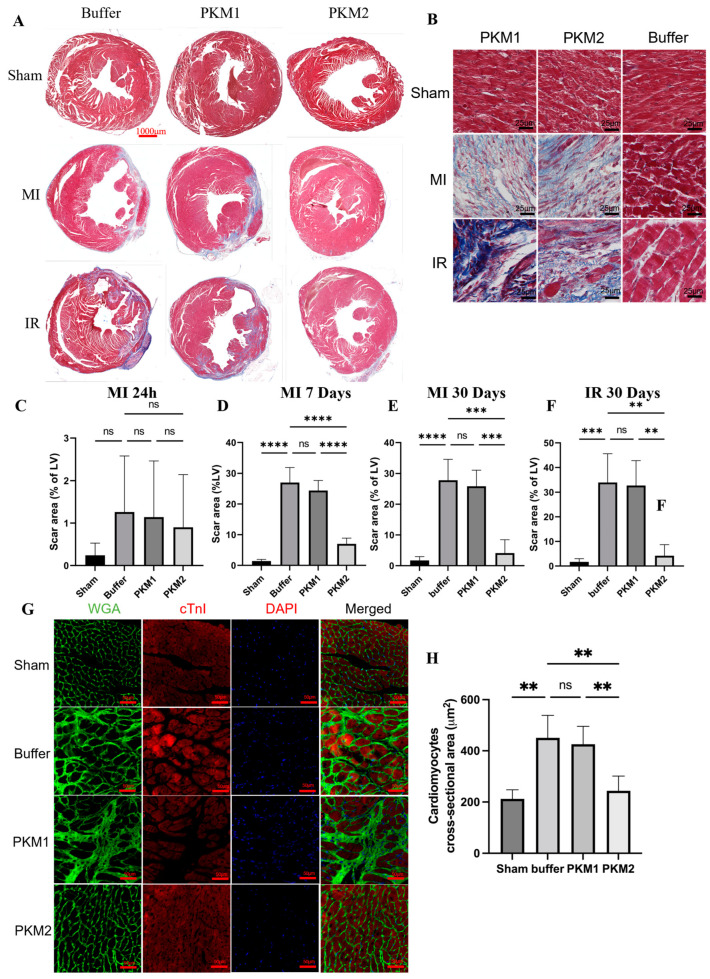
EcPKM2 decreases infarction scar sizes in the infarcted mouse model. (**A**,**B**) Representative images of Masson’s trichrome staining of the myocardium tissues (horizontal section) from infarcted mice 30 days after infarction induction. The images in (**B**) are magnified images to show details of myocardium structure. (**C**–**F**) Quantification of scar sizes of the infarcted hearts of mice that were treated with the indicated agents. (**C**) MI mice 24 h after infarction, (**D**) MI mice 7 days after infarction induction, (**E**) MI mice 30 days after infarction induction, and (**F**) IR mice 30 days after infarction. Quantities are presented as % of positive staining area in the whole left ventricle (LV), *n* = 5 unpaired *t*-test. (**G**) Representative magnified images WGA (green) or cTnI IF (red) staining of the myocardium tissues (horizontal section) of infarcted MI mice 30 days after infarction induction. (**H**) Quantification of the cardiomyocyte cross-sectional area of the infarcted MI hearts of mice that were treated with the indicated agents and 30 days after infarction induction. Quantities are presented as a cross-sectional area in μm, *n* = 5 unpaired *t*-test. Error bars in (**C**–**F**) represent mean ± S.E.M. Scale bars in (**A**), 1000 μm, in (**B**), 25 μm, and in (**G**), 50 μm. ns—statistical non-significance, ** *p* > 0.01, *** *p* > 0.005, **** *p* > 0.001.

**Figure 4 ijms-25-13246-f004:**
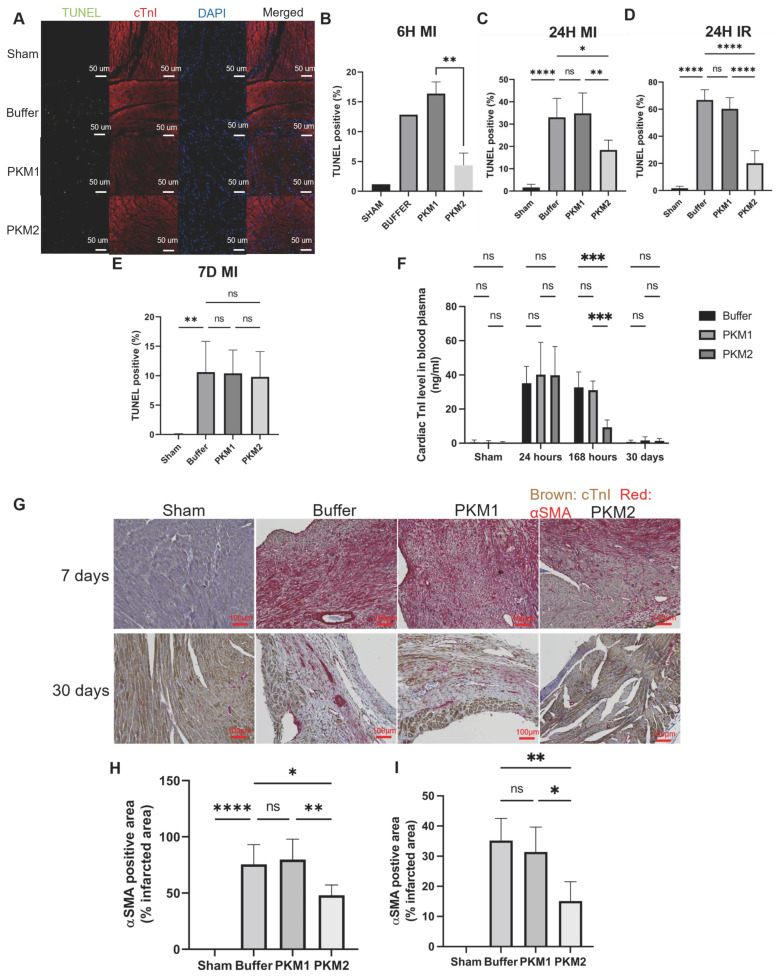
EcPKM2 protects cardiomyocytes from apoptosis and promotes proliferation in the myocardium tissues of infarcted hearts. (**A**) Representative images of TUNEL (green), cTnI (red), or DAPI (blue) staining of the myocardium tissues from 6 h after infarction induction of the mice. Overlays of the staining are merged. (**B**–**E**) Quantification of the TUNEL staining. The animals were euthanized 6 h (**B**), 24 h (**C**,**D**), or 7 days (**E**) after infarction (MI—(**B**,**C**,**E**) or IR—(**D**)). The quantity is presented as % of TUNEL positive area per view field (randomly selected 3 view fields per slide and 4 slides per animal), *n* = 6 unpaired *t*-test. The representative IF images of TUNEL and cTnI staining for C, D, and E are shown in Appendix A. (**F**) cTnI levels in plasma of infarcted mice were measured using cTnI ELISA assay. The animals had infarction induction by the indicated day and were treated with the indicated agents. The quantity is presented as ng/mL cTnI in plasma, *n* = 5 unpaired *t*-test. (**G**) Representative images of α-SMA IHC staining of the myocardium tissues of the infarction regions of infarcted mice. (**H**,**I**) Quantification of the α-SMA IHC staining. The quantity is presented as % of α-SMA positive area per view field (randomly selected 3 view fields per slide and 4 slides per animal), *n* = 7 unpaired *t*-test. The animals were euthanized 7 days (upper panel in (**G**,**H**)) or 30 days (lower panel in (**G**,**I**)) after MI infarction. The animals were treated with the indicated agents in (**G**–**I**). Error bars in (**B**,**D**–**F**,**H**,**I**) represent mean ± S.E.M. Scale bars in (**A**,**G**) 100 μm. ns—statistical non-significance, * *p* > 0.05, ** *p* > 0.01, *** *p* > 0.005, **** *p* > 0.001.

**Figure 5 ijms-25-13246-f005:**
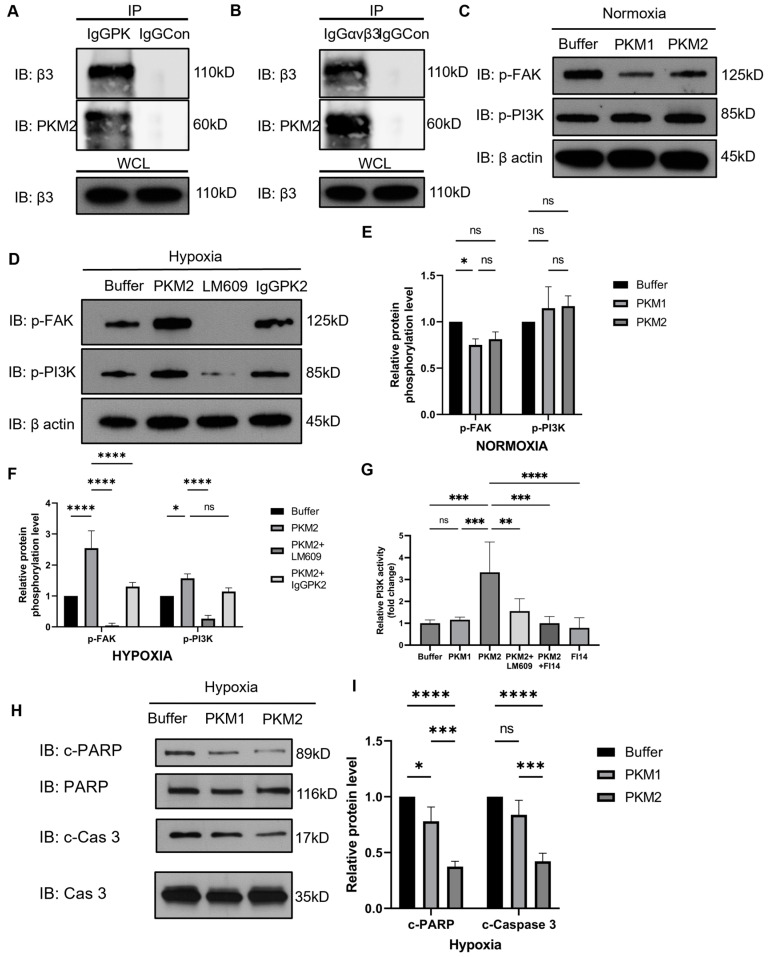
EcPKM2 interacts with integrin α_v_β_3_ and activates the integrin and downstream signaling. (**A**,**B**) Co-immunoprecipitation of G415R with integrin β3 in the extracts of human primary cardiomyocytes that were cultured under hypoxia conditions. The G415R was added into the cell extracts. Co-immunoprecipitations were carried out using anti-PKM2 antibody ((**A**), IgGPK) and anti-α_v_β_3_ ((**B**), IgGα_v_β_3_). IgGCon is the control for immunoprecipitation. The immunoblot of integrin β3 in whole cell extracts (WCL) is a control for the quantity of extracts used in Co-IP (Inputs). (**C**–**I**) Cellular levels of phosphorylated p-FAK and p-PI3K (**C**–**F**) and cleaved PARP (c-PARP) and caspase 3 (c-Cas 3) (**H**,**I**) in human primary cardiomyocytes as well as PI3K activity (**G**) that were cultured under normoxia (**C**,**E**) and hypoxia (**D**,**E**,**H**,**I**) conditions and treated with the indicated agents were analyzed by immunoblot (**C**,**D**,**H**) and three quantifications each of independent immunoblots (**E**,**F**,**I**), In (**C**,**D**,**H**), immunoblots of FAK, PI3K, PARP and caspase 3 are controls for total cellular levels of FAK, PI3K, PARP, and caspase 3, respectively. In (**E**,**F**,**I**), the quantity of IB protein level is presented as relative level by defining the measurements in buffer-treated cells as 1. In (**C**,**D**), immunoblot of β-actin is a loading control. The numbers on the sides of the immunoblots in (**A**–**D**,**H**) are molecular size markers. Error bars in (**E**,**F**) represent mean ± S.E.M. In (**E**–**G**,**I**), ns—statistical non-significance, * *p* > 0.05, ** *p* > 0.01, *** *p* > 0.005, **** *p* > 0.001 are calculated by an unpaired *t*-test.

**Figure 6 ijms-25-13246-f006:**
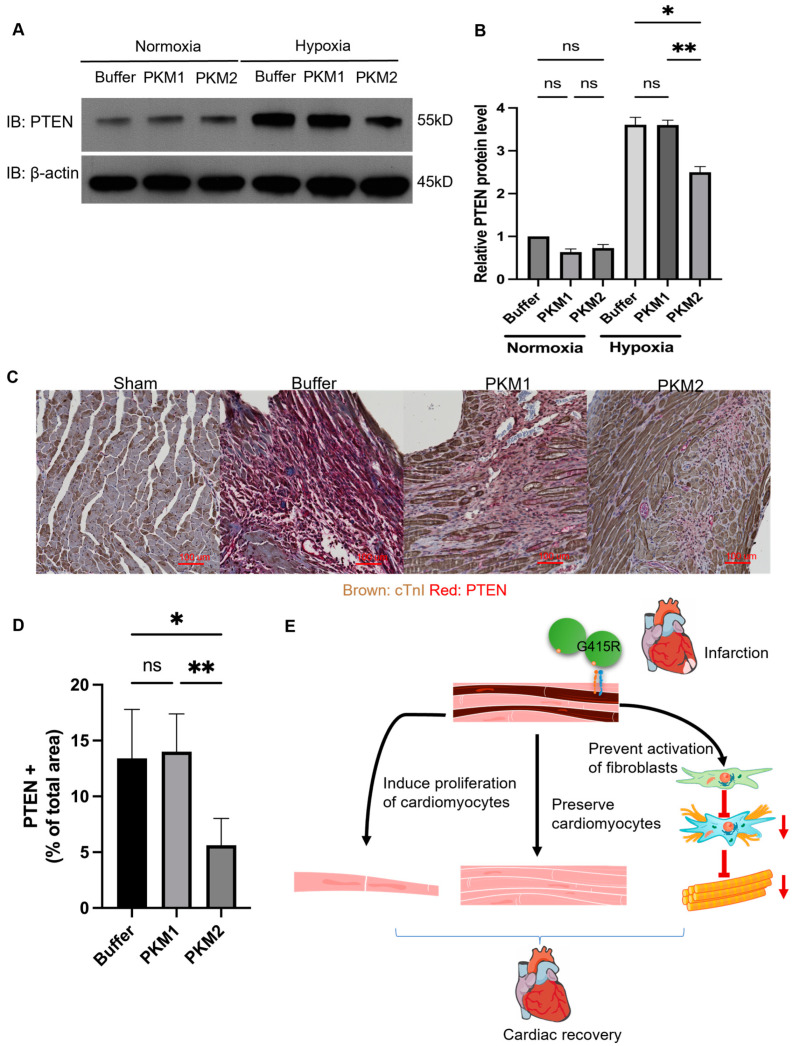
EcPKM2 regulates PTEN expression in cardiomyocytes in the myocardium tissues of infarcted hearts. (**A**,**B**) PTEN levels in the H9C2 cells cultured under hypoxia and normoxia conditions and treated with the indicated agents were analyzed by immunoblot (**A**) and four quantifications each of independent immunoblots (**B**) unpaired *t*-test. The quantity in (**B**) is presented as relative PTEN level by defining the PTEN level in buffer-treated cells as 1. Immunoblot of β-actin in (**A**) is a loading control. (**C**,**D**) Representative images (**C**) and quantifications (**D**) of PTEN IHC staining of the myocardium tissues of the infarction regions of infarcted mice (30 days post infarction induction) that were treated with the indicated agents. The quantity in (**D**) is presented as % of PTEN positive areas per field (randomly selected 3 view fields per slide and 4 slides per animal), *n* = 7 unpaired *t*-test. (**E**) Schematic diagram of the proposed mechanisms of EcPKM2 in preserving cardiomyocytes during myocardial infarction. The numbers on the sides of the immunoblots in A are molecular size markers. Error bars in (**B**,**D**) represent mean ± S.E.M. Scale bars in (**C**) are 100 μm. ns—statistical non-significance, * *p* > 0.05, ** *p* > 0.01.

## Data Availability

The data presented in this study are available on request from the corresponding author.

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
