# Peer review of "Extracellular PKM2 Preserves Cardiomyocytes and Reduces Cardiac Fibrosis During Myocardial Infarction"

_ijms, 2024, doi:10.3390/ijms252413246_

Round 1
Reviewer 1 Report
Comments and Suggestions for Authors
The authors have written a particularly interesting study about the protective functions of PKM2 in relation to myocardial infarction. This study performed well-designed experiments and well-explained results, which may contribute to the development of more effective therapies of the acute myocardial infarction in the future. This is a high quality manuscript, I do not have major criticism to make, however I would like to pose some questions and provide a few suggestions to the authors:
- Scales on microscopy images are not always clearly to be seen. In some cases, the coloring of certain columns is quite identical. I would recommend to use more different colors or different patterns for labeling different groups at column diagrams.
- At Figure 1. E, it is clearly visible that the PKM2 level in plasma does not significantly elevated after 24 hours of the myocardial infarction, but significantly does 168 hours after the infarction. I would be curious about the PKM2 plasma level between the two time points to exclude the possibility whether it peaks before 168h after the infarction.
- At Figure 2; it would probably make more sense to perform the data about the primary human cardiomyocytes and put the H9C2 data into the supplementary figures, as H9C2 cells often perform skeletal muscle phenotypic properties. At Figure 6; it was also probably more reasonable to show primary human cardiomyocyte data instead of H9C2.
- Heart weight vs body weight is a good option to evaluate the severity of cardiac hypertrophy. However, it would be better to perform the actual heart weights and body weights of the certain experimental groups, whereas their ratio can vary due to body weight changes resulted by a certain treatment.
- Heart weight vs body weight: it should be more precisely described what happened to these parameters. Is it due to the CMC hypertrophy or the body weight loss of the mice? What makes the difference between MI and IR?
- Could you explain in your manuscript how it is possible that in lung and liver, PMK2 facilitates fibrosis, but inhibits it in the heart?
- Whenever there are more than two groups receiving different treatments, wouldn’t it be more appropriate to use one-way ANOVA test than Student-test?
- The structuring of the text is sometimes confusing and illogical, in many places there is little connection between parts of the paragraphs, while the results in the diagrams do not follow the ordering of the paragraphs. More precise structuring would make the text easier to follow.
- There are some abbreviations that are not explained well (for example: ED, ES, EF). Additionally, the text needs some grammatical corrections.
- Probably, it should be explained better the effect of LM609.
Author Response
Comments 1: Scales on microscopy images are not always clearly to be seen. In some cases, the coloring of certain columns is quite identical. I would recommend to use more different colors or different patterns for labeling different groups at column diagrams.
Responses: We made some changes to make the scale bars as clear as possible. Thank the reviewers’ suggestion to use color bars/columns in our presentations. We used gray and black/white bars instead of colored columns. Gray and black/white bars presentation have its advantages.
Comment 2: At Figure 1. E, it is clearly visible that the PKM2 level in plasma does not significantly elevated after 24 hours of the myocardial infarction, but significantly does 168 hours after the infarction. I would be curious about the PKM2 plasma level between the two time points to exclude the possibility whether it peaks before 168h after the infarction.
Responses: The expression of PKM2 was not observed on day 3 after myocardial infarction (Please see Figure 1 B&C). Based on this observation and the previous report that PKM2 is only expressed in cardiomyocytes on day 4 after myocardial infarction (Ajit Magadum, et.al. 2020), the time point 168 hours (4 days) was selected.
Comment 3: At Figure 2; it would probably make more sense to perform the data about the primary human cardiomyocytes and put the H9C2 data into the supplementary figures, as H9C2 cells often perform skeletal muscle phenotypic properties. At Figure 6; it was also probably more reasonable to show primary human cardiomyocyte data instead of H9C2.
Responses: As suggested, we switched the human primary cardiomyocyte results to the main figure, and the results from H9C2 cells to supplementary figure.
Comment 4: Heart weight vs body weight is a good option to evaluate the severity of cardiac hypertrophy. However, it would be better to perform the actual heart weights and body weights of the certain experimental groups, whereas their ratio can vary due to body weight changes resulted by a certain treatment.
Response: As suggested, we replaced the HW/BW ratio by actual HW and BW (please see the new figure S3 E&F).
Comment 5: Heart weight vs body weight: it should be more precisely described what happened to these parameters. Is it due to the CMC hypertrophy or the body weight loss of the mice? What makes the difference between MI and IR?
Responses: As suggested, we replaced the HW/BW ratio by actual HW and BW (please see the new figure S3 E&F). It is clear that G415R treatment decreased heart weight compared to vehicle group. We do not know at this stage whether the weight changes were due to CMC hypertrophy or other factors. It is likely that CMC hypertrophy contributes to the differences. It seems that G415R had greater effects on MI compared to the IR group. One plausible
Comment 6: Could you explain in your manuscript how it is possible that in lung and liver, PMK2 facilitates fibrosis, but inhibits it in the heart?
Responses: The explanation is given in the last paragraph of the discussion section. Briefly, G415R treatment led to substantial less cardiomyocytes death, which consequently results in less cardiac fibroblasts activation (cardiomyocytes death is the main trigger for cardiac fibroblast activation in the myocardial infarction).
Comment 7: Whenever there are more than two groups receiving different treatments, wouldn’t it be more appropriate to use one-way ANOVA test than Student-test?
Responses: The statistical calculations most of the time were comparing two experimental groups. In this case, student t-test was used for statistical calculation. In a few cases, multiple experimental groups were compared. One-way ANOVA test was used in the statistical calculation.
Comment 8: The structuring of the text is sometimes confusing and illogical, in many places there is little connection between parts of the paragraphs, while the results in the diagrams do not follow the ordering of the paragraphs. More precise structuring would make the text easier to follow.
Responses: We did our best to re-organized the text to make our presentation clear. Thanks for reviewers’ comments/suggestion.
Comment 9: There are some abbreviations that are not explained well (for example: ED, ES, EF). Additionally, the text needs some grammatical corrections.
Responses: The abbreviations are clearly defined now (please see section 4.7. in the method section 4).
Comment 9: Probably, it should be explained better the effect of LM609.
Responses: The effect of LM609 has been added into the text (please see last paragraph in page 11).

Reviewer 2 Report
Comments and Suggestions for Authors
This is very interesting story. As we know, PKM2 is a key enzyme for TCA cycle. It is well-known for its intracellular role. However, its extracellular role is not confirmed. In this study, the authors firstly found it can be excreted, and then used recombinant PKM2 to investigate its effects on cardiocytes. It is a logical story. I have few concerns for this manuscript as followed.
1. PKM2 is expressed and released into extracellular space in myocardium tissue 4 days after myocardial infarction. My question for this point is that whether the release of cellular PKM2 was from the necrotic cells.
2. The resolution of figures are not good enough for review.
3. The molecular weight of WB bands should be added.
4. Tables 1-4 should not be listed in the main text. Please move them to supplemental data.
5. The authors have to correct some typoes and grammar errors.
Author Response
Comment 1: PKM2 is expressed and released into extracellular space in myocardium tissue 4 days after myocardial infarction. My question for this point is that whether the release of cellular PKM2 was from the necrotic cells.
Responses: It is not clear how PKM2 is released into extracellular space. It is possible that PKM2 is released from necrotic/dead cardiomyocytes. It is also possible that cardiomyocyte secrets PKM2.
Comment 2: The resolution of figures are not good enough for review.
Response: This may be due to effects of conversion to PDF format.
Comment 3: The molecular weight of WB bands should be added.
Responses: The molecular weight markers are added to the WB bands.
Comment 4: Tables 1-4 should not be listed in the main text. Please move them to supplemental data.
Responses: Table 1-4 in the material section has been moved to supplemental data.
Comment 5: The authors have to correct some typoes and grammar errors.
Responses: Typos and grammar errors are carefully corrected.

Round 2
Reviewer 2 Report
Comments and Suggestions for Authors
No further commnets.